# Variation in the Antibacterial and Antioxidant Activities of Essential Oils of Five New *Eucalyptus urophylla* S.T. Blake Clones in Thailand

**DOI:** 10.3390/molecules27030680

**Published:** 2022-01-20

**Authors:** Sapit Diloksumpun, Nalin Wongkattiya, Kittisak Buaban, Tharinee Saleepochn, Panawan Suttiarporn, Suwaporn Luangkamin

**Affiliations:** 1Department of Silviculture, Faculty of Forestry, Kasetsart University, Bangkok 10900, Thailand; sapit.d@ku.ac.th; 2Program in Biotechnology, Faculty of Science, Maejo University, Chiang Mai 50290, Thailand; nalin.wongkattiya@gmail.com (N.W.); kittisak.cue@gmail.com (K.B.); 3Department of Chemistry, Faculty of Science, Kasetsart University, Bangkok 10900, Thailand; fscitna@ku.ac.th; 4Faculty of Science, Energy and Environment, King Mongkut’s University of Technology North Bangkok, Rayong Campus, Rayong 21120, Thailand; panawan.s@sciee.kmutnb.ac.th; 5Department of Fundamental Science and Physical Education, Faculty of Science at Sriracha, Kasetsart University, Sriracha Campus, Chonburi 20230, Thailand

**Keywords:** *Eucalyptus urophylla*, *E. urophylla* × *E. camaldulensis* hybrid, *Eucalyptus* hybrid, essential oil, antibacterial, antioxidant, terpenoid

## Abstract

*Eucalyptus* oils are widely used for a variety of purposes. This study investigates the terpenoid compositions and antibacterial and antioxidant activities of eucalypt leaf oils extracted from four *E. urophylla* clones and one *E. urophylla* × *E. camaldulensis* hybrid clone grown in Thailand. According to GC/MS analysis, the *E. urophylla* oils were mainly composed of 1,8-cineole, *α*-terpinyl acetate, *β*-caryophyllene, and spathulenol, while 1,8-cineole, *α*-terpinyl acetate, *p*-cymene, and *γ*-terpinene were mostly identified in the hybrid oil. All eucalypt oils exhibited a significant bacteriostatic effect against Gram-positive bacteria, *Streptococcus pyogenes*, *Staphylococcus aureus*, *Listeria monocytogenes*, and *Bacillus cereus*. Only the hybrid oil had an effect on all Gram-negative bacteria tested, including *Salmonella typhi*, *Escherichia coli*, *Pseudomonas aeruginosa*, and *Enterobacter aerogenes*. These oils have antibacterial properties that vary according to their terpenoid content. Only the hybrid oil had a potent antioxidant effect, with an IC_50_ value of 4.21 ± 0.35 mg/mL for free radical (DPPH) scavenging. This oil’s antioxidant effect may be a result of the phenolic terpenoids, thymol and carvacrol. As a result, these oils may be a novel source of antibacterial and antioxidant agents. Additionally, the antibacterial and antioxidant capabilities of the *E. urophylla* × *E. camaldulensis* hybrid essential oil are reported for the first time.

## 1. Introduction

The *Eucalyptus* genus (eucalypt) is a tropical and subtropical tree in the Myrtaceae family. It consists of about 900 species, most originally from Australia [1]. Because it is fast growing, it has been broadly cultivated in many countries to utilize as wood for a diverse range of products. The eucalypt oils extracted from the leaves have excellent biological properties and are used in the pharmaceutical, agrochemical, cosmetic, and food industries [1,2,3]. *Eucalyptus* is one of Thailand’s most important economic woods for pulp and paper production. It accounts for 60% of annual commercial wood production and 56% of total commercial forest plantation area, excluding para rubber wood [4]. Two of the most promising *Eucalyptus* species widely planted in Thailand for commercial forest plantations are *E. urophylla* S.T. Blake (Timor white gum, Timor mountain gum), native to Indonesia, and *E. camaldulensis* Dehnh. (river red gum), originally from Australia [5,6]. To improve tree growth and wood quality for pulpwood and solid lumber processing, as well as pest and disease resistance, several tree breeding programs for these species have been developed in various regions, including crossbreeding between *E. urophylla* and *E. camaldulensis* in Thailand [7,8], to combine the desired genetic characteristics from the two different parents and to exploit heterosis, which is commonly referred to as specific crosses, and this gives individuals great advantages compared to the parent species [9]. Furthermore, the adoption of vegetative propagation to produce hybrid clones enables the capture of the total genetic variance in growth performance and wood properties and the production of a uniform raw material beneficial for industrial processes and product quality [9]. Heterosis in most hybrids between *E. urophylla* and *E. camaldulensis* has also been observed [9,10,11], and their promising clones have been commercially used [10,12]. Therefore, the possibility of utilizing the leaves and essential oils from the clonal plantations of *Eucalyptus*, which have found diverse applications in medicine, food, cosmetics, and agriculture, is worth investigating.

Hydrodistillation and steam distillation techniques are conventional techniques for isolating volatile terpenoids, which are important constituents in essential oils [13]. Due to their ease of operation and low cost, they are commonly used to extract essential oils from *Eucalyptus* leaves. The eucalypt leaf oils contain a variety of volatile monoterpenoids and sesquiterpenoids, the majority of which have a greater ratio of 1,8-cineole (eucalyptol). However, the dominant constituents in the oils, such as 1,8-cineole, *γ*-terpinene, *p*-cymene, *α*-pinene, spathulenol, and citronellal, vary according to the eucalypt species and may cause the oils to have different biological activities [1,3,14]. Eucalypt oils have been used in a variety of applications due to their ease of extraction and wide range of bioactivities. These oils have been shown to have antimicrobial, antiseptic, antiviral, antioxidant, anticancer, and anti-inflammatory activities. As a result, they are used in traditional herbal medicine (to treat respiratory diseases such as the common cold, flu, and sinus congestion), aromatherapy, and food preservation, and as active ingredients in cosmetics and household products [2,3]. Furthermore, the insecticidal, acaricidal, nematicidal, and herbicidal activities of eucalypt oils have been documented. Subsequently, they are also used in agriculture as natural pesticides, nematicides, herbicides, and insecticides [1,2,14].

Currently, natural products that have antioxidant and antimicrobial properties are blended in household products, toiletries, cosmetics, or topical applications to protect lipid peroxidation and microbial contaminations. Numerous studies on the antibacterial and antioxidant properties of eucalypt oils, including *E. urophylla* and *E. camaldulensis* leaf oil, have been published [1,2,3,15,16,17,18,19,20,21,22]. Interestingly, the *E. urophylla* and *E. camaldulensis* oils with dominant oxygenated terpenoids, such as 1,8-cineole, spathulenol, and carvacrol, have been shown to have significant antibacterial effects on a variety of pathogenic Gram-positive and Gram-negative bacteria [1,2,3,15,18,19,20,21]. Additionally, our group [22] has reported that the dominant phenolic terpenoids (2.18–7.25%), thymol and carvacrol, cause the *E. camaldulensis* oils to have potent antioxidant properties (DPPH radical scavenging, and IC_50_ values of 0.71–1.27 mg/mL), whereas the *E. urophylla* oil reported by Chahomchuen et al. [18] contained a small amount of thymol (0.13%), and was shown to have weak DPPH radical scavenging activity (IC_50_ 19.95 mg/mL). However, only a few studies of *E. urophylla* and *E. urophylla* × *E. camaldulensis* leaf oils have been performed [15,16,17,18,23,24,25,26,27,28,29]. Variations in the antibacterial and antioxidant activity of eucalypt oils, on the other hand, are highly dependent on the type and abundance of bioactive compounds, which vary depending on the genus species, genetic backgrounds, planting sites, and seasonal variations [1,3,20,21].

Therefore, the purpose of this research is to examine the phytochemical components, their antibacterial properties against eight pathogenic Gram-positive and Gram-negative bacterial strains, and the antioxidant properties of eucalypt oils extracted from the leaves of four *E. urophylla* clones and one *E. urophylla* × *E. camaldulensis* hybrid clone grown in Thailand’s Nakhon Ratchasima province. Our research is the first to examine the chemistry and bioactivities of pure and hybrid *E. urophylla* clones that have been registered as new clones in Thailand after being developed through a genetic improvement program by the Royal Forest Department (RFD) for high biomass production and good wood quality [30]. Moreover, such *Eucalyptus* clones providing essential oil as an important source of natural active substances may allow us to identify promising genotypes for both biomass production and the pharmaceutical, cosmetic, and household product industries.

## 2. Results and Discussion

### 2.1. Extraction of Eucalypt Oils

The clear yellowish eucalypt oils with their pleasant odor were obtained from the hydrodistillation of four *E. urophylla* clones (RFD 2–2, RFD 2–3, RFD 2–4, and RFD 2–6) and one hybrid clone of the *E. urophylla* × *E. camaldulensis* (RFD 2–5) leaves. The extraction yields of eucalypt oils varied between clones, ranging from 0.38 to 0.59% weight of oils per weight of dry leaves (Table 1). The hybrid clone (RFD 2–5) produced the highest yield, while the four *E. urophylla* clones contained less (0.38–0.54%). The eucalypt oil yielded by the RFD 2–6 clone was the highest among the *E. urophylla* clones. Regardless of clonal variations, the leaf essential oil yields of all the *E. urophylla* clones are related to those previously reported for the *E. urophylla* planted in the Congo [15], Lampang province, Thailand [18], Ethiopia [23], Brazil [24], and Côte d’Ivoire [27], ranging from 0.3 to 0.6%. Furthermore, the leaf essential oil yields of the *E. urophylla* and the *E. urophylla* × *E. camaldulensis* hybrid are higher than those reported by Li et al. [29], with 0.04 and 0.16%, respectively (Table 2). The genetic variations in *E. camaldulensis* leaf essential oil yields have also been reported by our group [22], ranging from 1.14 to 2.07%, and the oil yields were higher than *E. urophylla* oils. The essential oil yield from the leaves of this *Eucalyptus* hybrid clone was expected to be intermediate between the parental species.

### 2.2. GC/MS Analysis of Eucalypt Oils

The terpenoid compositions of five eucalypt leaf oils extracted from *E. urophylla* and *E. urophylla* × *E. camaldulensis* hybrid clones were tentatively identified by comparing their mass spectra with those contained in the NIST 14 mass spectral libraries and comparing the retention indices with those reported in the *NIST Chemistry WebBook*. The results are presented in Table 1. The relative concentration of all analytical compounds was determined by the percentage composition of each compound corresponding to the total compounds. Sixty-seven terpenoids were identified from the five eucalypt oils at different compositions, corresponding to 99–100% of the total oil compositions and including monoterpenoids (31.22–80.81%) and sesquiterpenoids (19.20–68.40%). These oils contain a high proportion of oxygenated compounds (55.26–77.19%).

Fifty-seven terpenoids were found in the clonal variation of four *E. urophylla* essential oils (RFD 2–2, RFD 2–3, RFD 2–4, and RFD 2–6), including monoterpenoids (31.22–68.95%) and sesquiterpenoids (31.08–68.40%). Only the essential oil of the RFD 2–4 clone contained more monoterpenes than sesquiterpenes. Among these oils, 61.62–77.19% were oxygenated terpenoids, with the greatest amount found in the essential oil of the RFD 2–6 clone, while only the essential oils of the RFD 2–3 and RFD 2–4 clones contained 0.48 and 1.08% phenolic terpenoids, respectively. Their *E. urophylla* oils were mainly composed of 1,8-cineole (16.34–22.75%), *α*-terpinyl acetate (4.79–7.72%), *β*-caryophyllene (3.82–11.79%), and spathulenol (2.05–11.93%). The greatest compositions of *β*-caryophyllene and spathulenol were found in the essential oils of RFD 2–3 and RFD 2–6 clones, respectively. In Table 2, the main terpenoid compositions of *E. urophylla* leaf essential oils compared with those of previous studies are shown. Previous studies have mostly found 1,8-cineole (eucalyptol) to be the main component in *E. urophylla* leaf essential oils [15,18,23,24,25,26,28,29], with the exception of Coffi et al. [27], who found 1,8-cineole to be a minor constituent in these oils.

The leaf essential oil of *E. urophylla* × *E. camaldulensis* hybrid clone (RFD 2–5) contained 39 terpenoids, including monoterpenoids (80.81%) and sesquiterpenoids (19.20%). There were 55.26% oxygenated terpenoids in this oil, with 3.45% phenolic terpenoids. This essential oil’s main constituents were 1,8-cineole (22.72%), *γ*-terpinene (17.99%), *p*-cymene (14.34%), and *α*-terpinyl acetate (8.66%). Previous studies by Li et al. [29] reported 1,8-cineole, *α*-pinene, and limonene to be major compounds in the hybrid oil (Table 2). Figure 1 illustrates the terpenoid structures of all the major compounds in the studied eucalypt oils. Previously, our group reported variations in the terpenoid compositions of *E. camaldulesis* leaf essential oils, which were mainly composed of *p*-cymene, 1,8-cineole, and *γ*-terpinene [22]. The terpenoid compositions and dominant constituents of hybrid eucalypt oil are expected to be a mixture of compounds derived from a cross between the parental species. The findings of the research indicated that the terpenoid components in eucalypt oils can vary significantly among clones of the pure species and between the pure and hybrid clones depending on the genetic background of the clones. Moreover, 1,8-cineole was found to be the most abundant in the compositions of all these eucalypt oils. Several studies show that eucalypt oils are greater in oxygenated terpenes, particularly 1,8-cineole, which seems to exhibit antimicrobial activity [1,3,31]. Furthermore, 1,8-cineole has been reported to have therapeutic potency for the treatment of numerous diseases, and no toxicity or carcinogenicity has been reported [32]. Additionally, the eucalypt oils of the clones RFD 2–3 and RFD 2–4, as well as the hybrid clone RFD 2–5, contained phenolic monoterpenes (Figure 1), the greatest amount of which was observed in the oil of the hybrid clone (3.45%). The phenolic terpenes have been shown to play an important role in the antioxidant effect of *E. camaldulensis* oils [22].

### 2.3. Antibacterial Activity of Eucalypt Oils

The results of an antibacterial agar disc diffusion assay on five *Eucalyptus* leaf oils against eight pathogenic bacterial strains are shown in Table 3. The inhibition zone diameter (IZD) was used to evaluate the antibacterial effect, which was classified as follows by Djabou et al. [33]: not sensitive (−) for diameters of ≤8 mm, moderately sensitive (+) for diameters between 8 and 14 mm, sensitive (++) for diameters between 14 and 20 mm, and extremely sensitive (+++) for diameters of ≥20 mm. In all bacterial tests except against *Bacillus cereus*, there were significant differences in antibacterial effects among the clones (*p* < 0.05). The findings also revealed that the four eucalypt oils of the *E. urophylla* clones (RFD 2–2, RFD 2–3, RFD 2–4, and RFD 2–6) had stronger antibacterial activity against Gram-positive bacteria than Gram-negative bacteria, whereas the eucalypt oils of the *E. urophylla* × *E. camaldulensis* hybrid clone (RFD 2–5) showed antibacterial effects against both Gram-positive and Gram-negative bacteria. All of these oils were extremely effective against *Streptococcus pyogenes*, with the oil from the RFD 2–2 clone exhibiting the strongest effect and being significantly different from the other oils, but not significantly different from tetracycline, the positive control. The eucalypt hybrid oil was more greatly effective against *Staphylococcus aureus* than all of the *E. urophylla* oils. These eucalypt oils also had an effect on *Listeria monocytogenes* and had a moderate effect against *Bacillus cereus*. The hybrid oil was also significantly more effective against Gram-negative bacteria, i.e., *Escherichia coli, Salmonella typhi*, *Pseudomonas aeruginosa*, and *Enterobacter aerogenes* than the *E. urophylla* oils, which were only moderately effective against *S. typhi* and inactive against *P. aeruginosa* and *E. aerogenes*. Additionally, the bacteriostatic effect of the hybrid oil against *P. aeruginosa* was considerably better than tetracycline. Nonetheless, all eucalypt oils were less effective against *S. aureus*, *B. cereus*, *L. monocytogenes*, *E. coli*, *S. typhi*, and *E. aerogenes* than the positive control, tetracycline, which is a board-spectrum antibiotic that has been widely used to treat a variety of clinical infections [34].

The minimum inhibitory concentration (MIC) and minimum bactericidal concentration (MBC) values of the *Eucalyptus* leaf oils on all test strains were also determined; these values ranged from < 0.06 to 16.00 and from 0.12 to 32 mg/mL, respectively (Table 4). According to Djabou et al. [33], the results were classified as follows based on the MIC and MBC values expressed in mg/mL: not sensitive (−) for values greater than 25.0 mg/mL, moderately sensitive (+) for values between 12.5 and 3.0 mg/mL, sensitive (++) for values between 2 and 0.4 mg/mL, and extremely sensitive (+++) for values equal or less than 0.2 mg/mL. The bacteriostatic effects of the tested eucalypt oils, as evaluated by the MIC and MBC, were related to IZD, with all the *E. urophylla* oils being more effective against Gram-positive bacteria than Gram-negative bacteria, while the hybrid eucalypt oil was effective against all tested strains. Additionally, the MIC values indicate that all the test oils were extremely sensitive to *S. pyogenes*, similar to the determination of inhibition zones, but the oil from the RFD 2–6 clone had the strongest effect. Based on MIC values, these oils had an effect on Gram-positive bacteria, along with *S. aureus*, *B. cereus*, and *L. monocytogenes*, with the oil from the RFD 2–6 clone having the greatest effect. In the case of Gram-negative bacterial strains, only the oil of the hybrid clone (RFD 2–5) was moderately sensitive to all the test strains, whereas the *E. urophylla* oils had a moderate effect only on *S. typhi*, and the other strains were inactive.

There have been a few studies on the antibacterial properties of *E. urophylla* leaf oil (Table 2). Only *S. aureus*, *E. coli*, and *P. aeruginosa* have been reported as bacteria tested [15,18]. However, the antibacterial properties of *E. urophylla* × *E. camaldulensis* hybrid leaf essential oil have not been reported elsewhere.

According to prior studies, Gram-positive bacteria are more susceptible to *Eucalyptus* oils than Gram-negative bacteria [1]. However, the variability in the antibacterial effects of eucalypt oils could be due to differences in the distribution of their major and minor constituents, the concentration of each component, or a synergetic effect between the constituents within the oils. The antibacterial action of oils is influenced by their hydrophilicity. Gram-positive bacteria have a cell wall composed of peptidoglycan linked with other molecules, such as protein or teichoic acid, while the cell walls of Gram-negative bacteria have an outer membrane made up of lipopolysaccharide (LPS), which acts as a barrier to hydrophilic molecules [35]. Gram-negative bacteria are therefore considered to be more resistant to the effects of essential oils that influence their hydrophilic groups (oxygenated compounds) than Gram-positive bacteria.

Correlations between the antibacterial effects of the five eucalypt oils and their constituents revealed that all essential oils containing high amounts of oxygenated terpenes (55.26–77.19%) were more effective against Gram-positive bacteria than Gram-negative bacteria. Many previous studies have found that eucalypt oils were higher in 1,8-cineole, which seems to have antibacterial properties [2,3,15,31,36,37]. However, the major oxygenated terpenoids in our essential oils, *α*-terpinyl acetate and spathulenol, as well as the minor constituents, borneol, citral, 4-terpineol, *α*-terpineol, thymol, carvacrol, and caryophyllene oxide, have been reported to have antibacterial effects [38,39,40,41,42]. Furthermore, the strongest activity of these oils against *S. pyogenes* was observed, and the oil of the *E. urophylla* clone (RFD 2–6), with the highest oxygenated terpenes, displayed the highest amount of activity (lowest MIC/MBC value). The findings supported previous reports that essential oils high in 1,8-cineole, *α*-terpinyl acetate, and spathulenol had a strong antibacterial effect against *S. pyogenes* [36,43,44]. Furthermore, the main monoterpene hydrocarbons, *p*-cymene and *γ*-terpinene, as well as the phenolic terpenoids, carvacrol and thymol, found in the oil of the *E. urophylla* × *E. camaldulensis* hybrid clone, may cause this oil to have bacteriostatic effects against both Gram-positive and Gram-negative bacteria. The antibacterial properties of these compounds, as well as the synergistic effect between carvacrol and cymene, have been reported [38,40,41,42,45].

### 2.4. Antioxidant Activity of Eucalypt Oils

The antioxidant activities of the clonal variations of four *E. urophylla* leaf oils and one *E. urophylla* × *E. camaldulensis* hybrid leaf oil were investigated using the DPPH free radical scavenging activity and the ferric reducing antioxidant power (FRAP) assay. With the DPPH assay, the five eucalypt oils were investigated in dose-dependent mode (0.5–5.0 mg/mL). With increasing concentrations of each eucalypt oil, there was an increase in free radical scavenging activity (%). Significant differences in antioxidant activity, i.e., DPPH inhibition, IC_50_, and the FRAP value, among the clones were observed (*p* < 0.05). The inhibition of these oils ranged from 22.56 ± 1.16 to 59.14 ± 1.66% based on the dose of 5.0 mg/mL. All eucalypt oils had IC_50_ values ranging from 4.21 ± 0.35 to 52.53 ± 19.96 mg/mL (Table 5). The activity of all tested oils was lower when compared to a standard synthetic antioxidant compound, such as BHT (IC_50_ value of 7.28 ± 0.47 μg/mL). The leaf oil of the *E. urophylla* × *E. camaldulensis* hybrid clone (RFD 2-5) had the highest free radical scavenging activity based on DPPH inhibition, which was significantly different from the other oils. Among the oils of the *E. urophylla* clones, RFD 2–3 and RFD 2–4 exhibited significantly greater DPPH radical scavenging activity compared with that of the RFD 2-2 and RFD 2-6 clones. The FRAP assay is described as a precise method to determine “antioxidant power,” with FRAP values calculated by correlating the absorbance change at 593 nm in sample mixtures to those comprising ferrous ions in a standard solution. The FRAP values of the five eucalypt oils ranged from 103.09 ± 3.47 to 337.86 ± 18.69 mM Fe (II)/g of oil (Table 5), and these are very low when compared to BHT, a synthetic antioxidant compound. The oil of the hybrid clone (RFD 2–5) had significantly higher FRAP values than the oils of the pure *E. urophylla* clones. In terms of the pure clone variation, the oil of the RFD 2–3 clone yielded the highest FRAP values related to the DPPH assay.

The antioxidant activities of the eucalypt oils from the four *E. urophylla* clones and the *E. urophylla* × *E. camaldulensis* hybrid clone planted in Thailand’s Nakhon Ratchasima province are reported here for the first time. However, there has yet to be a published report on the activity of the *E. urophylla* × *E. camaldulensis* hybrid oil, despite the fact that *E. urophylla* oil has been reported on by a few other researchers (Table 2). Furthermore, the antioxidant activities of the two *E. urophylla* oils from the RFD 2–3 and RFD 2–4 clones were greater than that of an *E. urophylla* oil cultivated in Northern Thailand (IC_50_ 19.95 mg/mL), reported by Chahomchuen et al. [18]. The difference in eucalypt oil activity could be attributed to differences in phytochemical compositions, which are influenced by genetic background, planting sites, plantation management, and climatic factors.

The phenolic terpenoids, thymol and carvacrol, are known to be antioxidant active compounds [22]. Our results revealed that the antioxidant property of the studied eucalypt oils was relative to the amount of thymol and carvacrol.

## 3. Materials and Methods

### 3.1. Plant Materials

Four *E. urophylla* clones and an *E. urophylla* × *E. camaldulensis* hybrid clone developed by the Royal Forest Department (RFD) were selected from a clonal test established in 2010 at the Northeast Forest Research and Development Center in Nakhon Ratchasima province, Thailand (14°28′ N, 101°54′ E) [30]. The *E. urophylla* clones (RFD 2–2, RFD 2–3, RFD 2–4, and RFD 2–6) were developed from a phenotypic selection of the half-sib families (open pollination), while the *E. urophylla* × *E. camaldulensis* hybrid clone (RFD 2–5) was developed from a phenotypic selection of the full-sib families (controlled pollination). Fresh mature leaves of all *Eucalyptus* clones were collected from planting sites in July 2019 for essential oil extraction.

### 3.2. Extraction of Eucalypt Oils

The dry leaves of the *Eucalyptus* samples were chopped into small pieces and ground with blenders after a week of air drying. The finely ground leaves (30 g) were hydrodistilled for 90 min in 250 mL of distilled water. After that, the distillate was extracted with methylene chloride and dried over sodium sulfate anhydrous before the solvent was drained away at low pressure. The eucalypt oil was then obtained and stored in the refrigerator until used. The oil yield was determined from the weight of the oil based on the weight of the dry leaves.

### 3.3. Analysis of Eucalypt Oils

The eucalypt oils were analyzed using a gas chromatography/mass spectrometer (GC/MS), namely, a Shimadzu GC-MS QP2020 with electron impact ionization (70 eV). One microliter of eucalypt oil solution (20 µL/mL) in dichloromethane was injected into a split/splitless inlet at 220 °C with a split ratio of 1:50. The carrier gas was helium, with a constant flow rate of 1 mL/min. Compounds were separated on an SH-Rxi-5Sil MS-fused silica capillary column (30 m length × 0.25 mm ID × 0.25 µm film thickness), with the temperature programmed to start at 60 °C, increase at 3 °C/min to 180 °C and then at 20 °C/min to 280 °C, and hold at 280 °C for 10 min (with a total run time of 60 min). The eluent was forwarded to the mass spectrometer via a transfer line maintained at 280 °C. The temperature of the ion source was 250 °C. The data were collected in scan mode (*m*/*z* range 45–550) with a 2 min solvent delay. The compounds were analyzed by correlating their mass spectra to those contained in the NIST 14 database. The relative amounts of compounds were calculated in percentages using a normalization method based on the peak area in the total peak chromatogram. The identification of compounds was also confirmed through a comparison of their relation indices relative to n-alkanes (C_7_–C_20_). The retention indices (RI) were calculated in accordance with a series of n-alkanes under the same chromatographic conditions according to Equation (1):(1)$$\mathrm{Retention}\;\mathrm{indices}\;\left(\mathrm{RI}\right)=\left[n+\left(\;\frac{tR\left(s\right)-tR\left(n\right)}{tR\left(N\right)-tR\left(n\right)}\right)\right]\times 100\;$$
where *n* is the number of carbon atoms, *t_R_*(*s*) is the retention time of the essential oil sample, *t_R_*(*n*) is the retention time of the smaller n-alkane (C_n_), and *t_R_*(*N*) is the retention time of the larger n-alkane (C_n+1_).

### 3.4. Antibacterial Activity

#### 3.4.1. Bacterial Strains

In this study, eight different bacterial strains were used. Four Gram-positive (*Staphylococcus aureus* DMST 8840, *Streptococcus pyogenes* DMST 30563, *Bacillus cereus* DMST 5040, and *Listeria monocytogenes* DMST 17303) and four Gram-negative (*Escherichia coli* DMST 4212, *Salmonella typhi* DMST 5784, *Pseudomonas aeruginosa* DMST 4739, and *Enterobacter aerogenes* DMST 8841) bacterial strains were received from Thailand’s Department of Medical Science, Ministry of Public health, Thailand. Before testing, they were subcultured for 24 h using brain heart infusion agar (BHA) at 37 °C.

#### 3.4.2. Determination of Inhibition Zones

The paper disc agar diffusion method was used to evaluate the eucalypt oil’s antibacterial activity [46]. The turbidity standard of 0.5 McFarland was used to adjust the bacterial suspension. Each bacterial suspension was spread onto Mueller–Hinton agar (MH) with a sterile cotton swab. A sample of 10 μL was dropped onto a disc with a diameter of 6 mm (Macherey-Nagel GmbH & Co., Dueren Germany). Before observing the inhibition zone, the disc was put on the inoculated BHA and incubated for 24 h at 37 °C. Tetracycline (Oxoid, UK) was used as a control. The inhibition zone diameter (IZD) was measured (mm) in triplicate.

#### 3.4.3. Determination of the Minimum Inhibitory Concentration (MIC) and Minimum Bactericidal Concentration (MBC)

The antibacterial effect of the eucalypt oils was also evaluated by determining the MIC and MBC in accordance with the CLSI, which are the antimicrobial disk susceptibility test standards [46]. In a 96-well microplate, the eucalypt oils were diluted two times to the desired concentrations. After adjusting each tested bacterium to the 0.5 McFarland standard turbidity, the bacterial suspension was diluted 100 times in brain heart infusion broth (BHB). After adding a diluted suspension of 50 μL to each well, the microplate was incubated for 24 h at 37 °C. Tetracycline was used as a positive control. The MIC was defined as the lowest concentration of essential oil that resulted in no visible bacterial growth. The MBC was carried out by culturing 10 μL of each clear well on BHA for 24 h at 37 °C. The MBC was defined as the lowest concentration of essential oil that showed no bacterial growth. This assay was performed in triplicate.

### 3.5. Antioxidant Activity

#### 3.5.1. DPPH Radical Scavenging Activity

The antioxidant properties of eucalypt oils were evaluated using the 2,2-diphenyl-1-picrylhydrazyl (DPPH) radical scavenging activity, as previously described [22]. The methanolic solutions of essential oils (1 mL), ranging in concentration from 1 to 10 mg/mL, were combined with 2 mL of DPPH methanolic solution (0.1 mM). The mixtures were incubated for 30 min in a cabinet at room temperature before the absorbance was measured at 517 nm against a blank. As a positive control, butylated hydroxytoluene (BHT) was used. Equation (2) was used to calculate the % DPPH of the radicals’ inhibition:DPPH Inhibition (%) = (A_b_ − A_o_/A_b_) × 100(2)
where A_b_ defines the blank’s absorbance and A_o_ defines the essential oil’s absorbance. The IC_50_, represented as the oil concentration that inhibits free radicals by 50%, was determined using Probit analysis at a 95% confidence level [47].

#### 3.5.2. Ferric Reducing Antioxidant Power (FRAP)

The FRAP assay was determined using a method previously described [22]. The stock solutions consisted of an acetate buffer (300 mM, pH 3.6), a 2,4,6-tri(2-pyridyl)-s-triazine (TPTZ) solution (10 mM) in hydrogen chloride (40 mM), and ferric chloride (20 mM). The fresh FRAP solution was prepared by combining the acetate buffer, TPTZ, and ferric chloride solutions in a 10:1:1 ratio. The eucalypt oils (100 µL) were combined with the FRAP solution (900 µL) and left to react in the dark for 30 min at 37 °C. The resulting ferrous tripyridyl triazine complex was colorimetrically measured at 593 nm. The FRAP value was determined using the calibration curve of ferrous sulfate standard solutions (0.05–1 mM) and expressed as milli-molars of Fe (II) per gram of oil. The synthetic antioxidant, BHT, was also analyzed for comparison. The analyses were performed in three replicates, with the average value calculated in each case.

### 3.6. Statistical Analysis

The results of the antibacterial and antioxidant activity tests were represented as mean ± SD. The analysis of variance (ANOVA) procedure was used to investigate clonal variation in the antibacterial and antioxidant activities, and Duncan’s multiple range test, a post hoc test, was performed to determine any significant differences (*p* < 0.05) between clones. The Pearson test was used to analyze the correlation between the two variants. For the statistical analyses, SPSS software (SPSS v.26 for Windows; IBM Crop., Armonk, NY, USA) was used.

## 4. Conclusions

Five eucalypt oils were hydrodistilled from the leaves of four *E. urophylla* clones (RFD 2–2, RFD 2–3, RFD 2–4, and RFD 2–6) and an *E. urophylla* × *E. camaldulensis* hybrid clone (RFD 2–5). The clear and yellowish eucalypt oils were obtained in a 0.38–0.59% yield on a dry leaf basis, with the oil yield of the hybrid clone being higher than the pure *E. urophylla* clones. Based on their chemical compositions, the eucalypt oils obtained in this study were qualitatively and quantitatively showed differences among the *E. urophylla* clones and hybrid clone. The *E. urophylla* oils mainly comprised 1,8-cineole, *α*-terpinyl acetate, *β*-caryophyllene, and spathulenol, whereas a mixture of compounds from a cross between the parental species made up the major and minor compositions of eucalypt hybrid oil. The antibacterial activities of these oils were investigated, and it was found that all *E. urophylla* oils were more potent against Gram-positive bacterial strains than Gram-negative bacterial strains, but a eucalypt hybrid oil was effective against all strains tested. Furthermore, *S. pyogenes* was extremely sensitive to these oils. The oxygenated terpenes, particularly 1,8-cineole, *α*-terpinyl acetate, and spathulenol, may be responsible for these eucalypt oils’ strong antibacterial effect against Gram-positive bacteria. The oils of the hybrid clone could also inhibit all Gram-negative bacteria, most probably due to the dominance of monoterpenes, *p*-cymene, and *γ*-terpinene, or to a synergistic effect of these and phenolic terpenoids. All eucalypt oils were tested for antioxidant activities, and essential oil from the hybrid eucalypt clone was observed to be more potent than the pure *E. urophylla* clones. Phenolic terpenoids, such as thymol and carvacrol, may contribute to the antioxidant properties of these oils. Our findings are the first to describe the antibacterial and antioxidant activities of the *E. urophylla* × *E. camaldulensis* hybrid’s leaf essential oil.

According to the study’s findings, all of the investigated eucalypt oils might be used as a new significant source of natural bioactive agents for healthcare, beauty care, and household products.

## Figures and Tables

**Figure 1 molecules-27-00680-f001:**
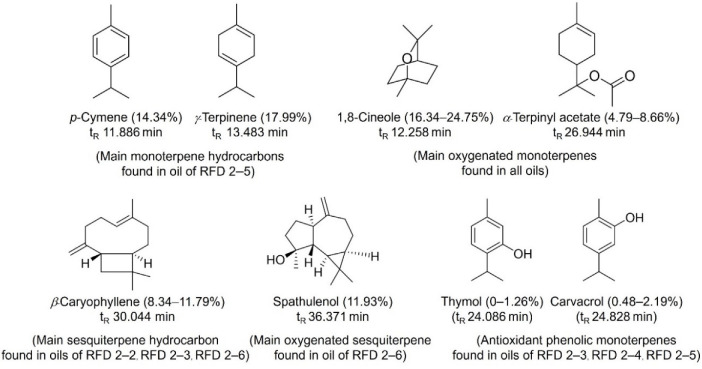
Chemical structures of main terpenoids and phenolic terpenoids found in essential oils of the *E. urophylla* clones (RFD 2–2, RFD 2–3, RFD 2–4, and RFD 2–6) and the *E. urophylla* × *E. camaldulensis* hybrid clone (RFD 2–5) that are important to the bioactivity of oils.

**Table 1 molecules-27-00680-t001:** Extraction yields and terpenoids compositions (%) of the five eucalypt oils from the leaves of *E. urophylla* clones (RFD 2–2, RFD 2–3, RFD 2–4, and RFD 2–6) and the *E. urophylla* × *E. camaldulensis* hybrid clone (RFD 2–5) by GC/MS.

No.	Compound ^a^	t_R_ ^b^	RI ^c^	Composition (%) ^d^				
				RFD 2–2	RFD 2–3	RFD 2–4	RFD 2–6	RFD 2–5
1	*α*-Pinene	7.918	959	2.93	2.62	6.47	0.82	1.26
2	*p*-Cymene	11.886	1043	-	2.79	7.10	0.74	14.34
3	D-Limonene	12.117	1048	1.67	1.31	2.28	1.43	3.16
4	1,8-Cineole	12.258	1051	22.75	16.34	24.75	21.07	22.72
5	*trans*-*β*-Ocimene	12.446	1055	-	-	-	-	1.98
6	*γ*-Terpinene	13.483	1077	-	-	-	0.24	17.99
7	Terpinolene	14.774	1105	-	-	-	-	0.55
8	*trans*-Linalool oxide (furanoid)	14.815	1105	-	-	0.91	-	-
9	Fenchol	16.347	1138	0.50	-	1.25	0.26	-
10	*trans*-Pinocarveol	17.372	1160	0.77	0.49	4.04	-	-
11	Pinocarvone	18.389	1181	0.32	-	1.78	-	-
12	Borneol	18.869	1192	1.08	-	2.04	0.39	-
13	Citral	19.205	1199	-	-	-	-	0.61
14	4-Terpineol	19.303	1201	-	0.54	0.58	-	2.54
15	*p*-Cymen-8-ol	19.605	1207	-	-	-	-	0.27
16	*trans*-*p*-Mentha-1(7),8-dien-2-ol	19.649	1208	-	-	0.96	-	-
17	*α*-Terpineol	20.022	1216	3.23	1.86	5.56	2.13	2.10
18	*cis*-Sabinol	20.840	1234	-	-	-	-	0.35
19	*cis*-Carveol	21.123	1240	-	-	0.34	-	-
20	*cis*-*p*-Mentha-1(7),8-dien-2-ol	21.593	1250	-	-	0.97	-	-
21	Carvotanacetone	22.465	1268	-	-	0.56	-	-
22	Piperitone	22.696	1273	-	-	0.88	-	0.23
23	*trans*-Linalool oxide acetate (pyranoid)	23.993	1299	-	-	0.38	-	-
24	Thymol	24.086	1303	-	-	-	-	1.26
25	Carvacrol	24.828	1318	-	0.48	1.08	-	2.19
26	*p*-Menth-4(8)-en-2,5-diol	25.448	1332	-	-	-	-	0.25
27	*α*-Terpinyl acetate	26.944	1363	7.72	4.79	7.02	5.56	8.66
28	*α*-Copaene	28.200	1388	0.93	0.83	-	0.58	0.35
29	Geranyl acetate	28.329	1392	-	-	-	0.43	0.35
30	*α*-Gurjunene	29.547	1423	0.35	0.56	-	-	-
31	*β*-Caryophyllene	30.044	1435	8.38	11.79	3.82	8.34	0.37
32	Aromadendrene	30.836	1455	-	0.33	-	-	1.42
33	Cadina-3,5-diene	31.275	1466	0.30	-	-	-	-
34	Humulene	31.530	1473	1.47	2.18	0.66	1.38	-
35	Alloaromadendrene	31.720	1478	0.62	1.19	0.49	1.18	0.35
36	*β*-Cadinene	32.240	1491	0.75	0.59	-	-	-
37	*γ*-Selinene	32.883	1507	-	0.62	-	0.44	-
38	Viridiflorene	33.015	1510	-	-	-	-	0.25
39	*γ*-Muurolene	33.023	1511	1.50	1.71	-	0.34	-
40	Bicyclogermacrene	33.190	1515	1.36	2.29	0.76	0.51	0.26
41	*α*-Muurolene	33.335	1519	0.74	0.59	-	0.42	-
42	*δ*-Cadinene	34.122	1539	4.83	4.06	1.22	1.96	1.59
43	*trans*-Calamenene	34.236	1541	2.62	2.23	1.20	3.19	0.43
44	Zonarene	34.302	1543	1.88	1.92	0.58	-	-
45	Cubenene	34.666	1552	0.61	-	-	-	0.45
46	*α*-Dehydro-ar-himachalene	34.765	1555	0.28	-	-	0.27	-
47	*α*-Calacorene	34.987	1561	1.21	0.39	-	0.69	-
48	Epiglobulol	35.811	1581	-	0.45	-	-	0.34
49	Maaliol	36.105	1588	0.86	1.87	1.00	1.32	-
50	Spathulenol	36.371	1596	2.05	3.78	3.35	11.93	1.34
51	Caryophyllene oxide	36.563	1600	4.02	2.48	1.54	6.57	0.44
52	Globulol	36.721	1604	4.80	5.70	3.84	5.11	3.49
53	Viridiflorol	37.041	1613	2.40	5.01	3.12	3.94	0.45
54	Cubeban-11-ol	37.142	1615	1.15	2.71	1.53	1.60	0.30
55	Ledol	37.445	1623	0.89	1.04	0.53	1.10	-
56	Rosifoliol	37.530	1625	0.71	2.00	0.89	1.13	-
57	Humulene oxide II	37.625	1627	0.49	-	-	0.85	-
58	*α*-Eudesmol	38.180	1641	0.84	2.43	1.32	1.56	0.34
59	1,10-Di-epi-cubenol	38.333	1645	4.61	3.63	2.19	4.10	3.67
60	*γ*-Eudesmol	38.494	1649	-	-	-	-	0.48
61	Caryophylladienol II	38.655	1654	-	-	-	1.15	-
62	Epicubenol	38.917	1660	3.91	3.06	1.31	2.98	0.72
63	*δ*-Cadinol	39.041	1663	1.87	1.34	0.60	1.52	0.27
64	*α*-Copaene-11-ol	39.155	1666	-	-	-	0.26	-
65	*β*-Eudesmol	39.315	1670	-	-	-	-	1.89
66	*α*-Cadinol	39.361	1671	1.08	1.00	1.13	1.74	-
67	Neointermedeol	39.444	1673	0.59	0.62	-	0.49	-
	Total terpenoids			99.07	99.62	100	99.72	100
	Total monoterpenoids			40.97	31.22	68.95	33.07	80.81
	Total sesquiterpenoids			58.10	68.40	31.08	66.65	19.20
	Oxygenated terpenoids			66.64	61.62	75.45	77.19	55.26
	Phenolic terpenoids			0	0.48	1.08	0	3.45
	Extraction yield ^e^ (%*w*/*w*)			0.48	0.39	0.38	0.54	0.59

^a^ Compound was tentatively identified by comparing it with mass spectrum data from the NIST library. ^b^ Retention time (t_R_) in minutes. ^c^ Retention indices (RI) in relation to n-alkanes (C_7_–C_20_) under the same conditions. ^d^ % composition was the relative amount in all analyzed compounds, calculated from peak area. -: Not detected.

**Table 2 molecules-27-00680-t002:** Comparison of essential oil yields, main terpenoids, and antibacterial and antioxidant activities of *E. urophylla* leaf oils with some those of previous studies.

Source	Oil Yield (%*w*/*i*)	Main Terpenoids (% Compositions)	Antibacterial Activity (IZD mm)	Antioxidant Activity	Author
Thailand (*E. urophylla*, RFD clones)	0.39–0.54	1,8-Cineole (**16**–**25**) *α*-Terpinyl acetate (**5**–**8**) *β*-Caryophyllene (**4**–**12**) Spathulenol (**2**–**12**)	*Staphylococcus aureus* (**16**) *Streptococcus pyogenes* (**19**–**32**) *Bacillus cereus* (**12**,**13**) *Listeria monocytogenes* (**17**–**19**)	IC_50_ (DPPH) 9.42–12.05 mg/mL (RFD 2–3 and RFD 2–4)	This study
Thailand (*E. urophylla* × *E. camaldulensis*, RFD clones)	0.59	1,8-Cineole (**23**) *γ*-Terpinene (**18**) *p*-Cymene (**14**) *α*-Terpinyl acetate (**9**)	*Staphylococcus aureus* (**24**) *Streptococcus pyogenes* (**19**) *Bacillus cereus* (**13**) *Listeria monocytogenes* (**16**) *Escherichia coli* (**12**) *Salmonella typhi* (**18**) *Pseudomonas aeruginosa* (**14**) *Enterobacter aerogenes* (**13**)	IC_50_ (DPPH) 4.21 mg/mL	This study
Lampang, Thailand (*E. urophylla)*	0.6	1,8-Cineole (**43**) *γ*-Terpinene (**27**)	*Staphylococcus aureus* (**16**) *Bacillus subtilis* (**16**) *Escherichia coli* (**12**) *Pseudomonas aeruginosa* (**30**) *Staphylococcus intermedius* (**17**)	IC_50_ (DPPH) 19.95 mg/mL	Chahomchuen et al., 2020 [18]
Portugal (*E. urophylla*)	0.86	*α*-Phellandrene (**45**) 1,8-Cineole (**23**)	*-*	TEAC (ABTS) 0.63 μmol TE/g EO [17]	Faria et al., 2013 [28], Miguel et al., 2018 [17]
Brazil (*E. urophylla)*	0.29	1,8-Cineole (**53**) *α*-Pinene (**8**)	*Salmonella enteritidis* (**7**) [16]	-	Batista-Pereira et al., 2006 [24], Ambrosio et al., 2017 [16]
Congo (*E. urophylla)*	0.53	1,8-Cineole (**58**) *α*-Pinene (**10**)	*Bacillus subtilis* (**16**) *Escherichia coli* (**7**–**13**) *Klebsiella oxytoca* (**15**) *Klebsiella pneumoniae* (**17**) *Proteus vulgaris* (**12**)	-	Cimanga et al., 2002 [15]
China (*E. urophylla*)	0.04	1,8-Cineole (**63**,**64**) *α*-Pinene (**11**–**23**) *α*-Terpinyl acetate (**8**)	*-*	-	Li et al., 2020 [29]
China (*E. urophylla* × *E. camaldulensis*)	0.16	1,8-Cineole (**44**) Limonene (**27**) *α*-Pinene (**16**)	*-*	-	Li et al., 2020 [29]
Côte d’Ivoire (*E. urophylla*)	0.4	*γ*-Terpinene (**23**) *p*-Cymene (**17**) *β*-Pinene (**15**) *α*–Phellandrene (**9**)	*-*	-	Coffi et al., 2012 [27]
Taiwan (*E. urophylla*)	2.2	1,8-Cineole (**58**) *α*-Terpinyl acetate (**15**)	-	-	Cheng et al., 2009 [25]
Ethiopia (*E. urophylla*)	0.4	1,8-cineole (**34**) *α*-Pinene (**13**) *α*-Terpinyl acetate (**12**) Limonene (**10**)	*-*	-	Dagne et al., 2000 [23]

IZD: Inhibition zone diameter. -: no reported.

**Table 3 molecules-27-00680-t003:** The inhibition zone diameters (IZDs) of five eucalypt oil against eight bacterial strains.

Bacterial Strains	IZD (mm) ^a^					
	RFD 2–2	RFD 2–3	RFD 2–4	RFD 2–6	RFD 2–5	Tetracycline ^b^
Gram positive						
*Staphylococcus aureus* (DMST 8840)	16.42 ± 0.78c	15.54 ± 1.45c	15.77 ± 0.37c	16.09 ± 2.09c	24.23 ± 2.27b	28.33 ± 2.99a
*Streptococcus pyogenes* (DMST 30563)	32.49 ± 4.61a	18.87 ± 0.94b	21.26 ± 2.46b	21.41 ± 1.29b	18.77 ± 1.49b	34.30 ± 2.95a
*Bacillus cereus* (DMST 5040)	13.43 ± 2.11	13.29 ± 4.65	12.89 ± 2.54	11.69 ± 5.09	12.59 ± 2.94	18.23 ± 1.59
*Listeria monocytogenes* (DMST 17303)	19.49 ± 2.13b	18.26 ± 1.90bc	17.33 ± 1.03bc	17.85 ± 0.89bc	15.85 ± 1.35c	22.76 ± 1.13a
Gram negative						
*Escherichia coli* (DMST 4212)	6.00 ± 0.00d	6.00 ± 0.00d	8.36 ± 0.12c	6.00 ± 0.00d	11.75 ± 1.10b	23.95 ± 1.60a
*Salmonella typh**i* (DMST 5784)	7.87 ± 1.72c	8.35 ± 2.06c	10.78 ± 1.60c	8.28 ± 2.05c	17.97 ± 2.47b	28.17 ± 2.68a
*Pseudomonas aeruginosa* (DMST 4739)	6.00 ± 0.00b	6.00 ± 0.00b	6.00 ± 0.00b	6.00 ± 00b	14.41 ± 5.14a	8.53 ± 1.10b
*Enterobacter aerogenes* (DMST 8841)	6.00 ± 0.00d	6.00 ± 0.00d	7.71 ± 0.61c	6.00 ± 0.00d	12.69 ± 2.07b	17.79 ± 0.37a

^a^ IZD includes the diameter of the disc (6 mm). ^b^ Tetracycline was used as a positive control. RFD 2-2, RFD 2-3, RFD 2-4, and RFD 2-6 are *E. urophylla* oils. RFD 2-5 is the hybrid *E. urophylla* x *E. camaldulensis* oil. Data are represented as the mean ± SD in triplicate, and different lowercase letters in the same row represent significant difference among clones by Duncan’s multiple range test (*p* < 0.05).

**Table 4 molecules-27-00680-t004:** The MIC and MBC of five eucalypt oils against eight bacterial strains.

Bacterial Strains	MIC/MBC (mg/mL)					
	RFD 2-2	RFD 2-3	RFD 2-4	RFD 2-6	RFD 2-5	Tetracycline ^a^
Gram positive						
*Staphylococcus aureus* (DMST 8840)	1.0/4.0	1.0/8.0	2.0/8.0	0.5/4.0	2.0/8.0	<0.0002/0.0019
*Streptococcus pyogenes* (DMST 30563)	0.12/0.5	0.12/0.25	0.12/0.5	<0.06/0.12	0.25/0.5	<0.0002/0.0019
*Bacillus cereus* (DMST 5040)	0.5/2.0	0.5/4.0	0.5/4.0	0.12/1.0	1.0/4.0	0.0019/0.0156
*Listeria monocytogenes* (DMST 17303)	0.5/1.0	0.5/1.0	0.5/2.0	0.5/0.5	1.0/4.0	0.0009/0.0078
Gram negative						
*Escherichia coli* (DMST 4212)	16.0/16.0	16.0/16.0	8.0/16.0	16.0/32.0	8.0/8.0	0.0009/0.0078
*Salmonella typhi* (DMST 5784)	4.0/8.0	4.0/8.0	4.0/8.0	4.0/8.0	4.0/4.0	0.0009/0.0078
*Pseudomonas aeruginosa* (DMST 4739)	8.0/16.0	8.0/16.0	8.0/16.0	8.0/16.0	4.0/4.0	0.06/>0.25
*Enterobacter aerogenes* (DMST 8841)	16.0/16.0	16.0/16.0	16.0/16.0	16.0/16.0	8.0/8.0	0.0078/0.125

^a^ Tetracycline was used as a positive control. RFD 2-2, RFD 2-3, RFD 2-4, and RFD 2-6 are *E. urophylla* oils. RFD 2-5 is the hybrid *E. urophylla* × *E. camaldulensis* oil.

**Table 5 molecules-27-00680-t005:** Antioxidant activities of the five eucalypt oils.

Eucalypt Oils ^a^	Antioxidant Activities		
	% DPPH Inhibition ^b^	IC_50_ (mg/mL)	FRAP (mM Fe (II)/g of Oil)
RFD 2-2	28.08 ± 0.32c	52.53 ± 19.96a	129.38 ± 5.65c
RFD 2-3	36.66 ± 1.22b	9.42 ± 0.57b	230.15 ± 66.14b
RFD 2-4	36.48 ± 0.50b	12.05 ± 0.51b	163.96 ± 4.03bc
RFD 2-6	22.56 ± 1.16d	51.59 ± 27.06a	103.09 ± 3.47c
RFD 2-5	59.14 ± 1.66a	4.21 ± 0.35b	337.86 ± 18.69a

^a^ Leaf eucalypt oils of the *E. urophylla* clones (RFD 2–2, RFD 2–3, RFD 2–4, and RFD 2–6) and the *E. urophylla* × *E. camaldulensis* hybrid clone (RFD 2–5). ^b^ % DPPH inhibition at 5 mg/mL of eucalypt oils. As a positive control, BHT was used with an IC_50_ of 7.28 ± 0.47 μg/mL. Data are represented as mean ± SD in triplicate, and different lowercase letters in the same column represent significant difference among clones according to Duncan’s multiple range test (*p* < 0.05).

## Data Availability

Data are contained within the article.

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
