# Peer review of "Variation in the Antibacterial and Antioxidant Activities of Essential Oils of Five New Eucalyptus urophylla S.T. Blake Clones in Thailand"

_molecules, 2022, doi:10.3390/molecules27030680_

Round 1

Reviewer 1 Report

Dear Editor,

Overall, this article can be considered for publication in Molecules: An International Journal, seen from data updates to the discovery of new drugs from those obtained from hybrid clones and combinations. Besides being able to provide new information regarding the learning model of hybrid clones and their combinations, this can also be a consideration for researchers in developing new drugs, especially essential oils Overall, this manuscript needs to be reviewed because it lacks clarity of novelty. All parts of the manuscript should be majorly corrected as I wrote in the comments. It need to be emphasized and improved, such as the following suggestions:

  1. Introduction (Page 2, Line 50): says “Furthermore, the adoption of vegetative propagation to produce hybrid clones enables the capture of the total genetic variance in growth performance and wood properties and the production of a uniform raw material benefit to industrial process and product quality”, even though the basis for this recognition is not clear and should be followed by the reason that the hybrid clones have shown very promising results.
  2. Introduction: To bring up a novelty, it would be better if a literature search related to previous research was carried out so that there would be a comparison between previous research and the research that had been done.
  3. Results and Discussion (table 1, line 161): there is no data related to the retention time of each compound and the difference between all samples used.
  4. Results and discussion (Figure 1, Line 173): it is not explained which compound is attached at what retention time and what its use is so that this research is considered important.
  5. Results and Discussion (line 164): there is a description of the use of c-alkanes but no explanation regarding the purpose of using c-alkanes.
  6. Results and Discussion: There is no elaboration on the results of the statistical analysis.
  7. Method (Line 358): mentioned using tetracycline as a control but there is no explanation of the comparison of the results.
  8. Method (Line 382): ​​BHT is mentioned as a control but there is no explanation for the comparison of the results.
  9. Since this manuscript has many abbreviations, it would be better if a list of abbreviations was added

Reviewer 2 Report

The manuscript by Sapit Diloksumpun et al., describes the terpenoid composition and antibacterial and antioxidant activities of eucalypt leaf oils extracted from four E. urophylla clones and one E. urophylla x E. camaldulensis hybrid clone grown in Thailand. Several monoterpenes and sesquiterpenes were tentatively identified by GC-MS. Antibacterial activity was qualitatively and quantitatively determined against gram-positive and gram-negative bacteria. The antioxidant activity was assessed by the DPPH and FRAP assays.

The manuscript falls in the scope of Molecules, however, a careful revision is mandatory because there are several points that have to be improved and corrected.

Abstract

Lines 25-26: “Only hybrid oil demonstrated resistance to all Gram-negative bacteria…”; this sentence must be corrected because an oil can´t be resistant to bacteria.

Introduction

Lines 59-61: this sentence needs an explanation. What may have different biological activities? The oil or the isolated compounds? Please re-write.

Lines 63-64: this sentence must be re-written. The oils have antimicrobial, antiseptic, antiviral (…) activities. What about “chemotherapy agent” or “respiratory treatment properties” – these are not biological activities.

Lines 67-70: needs revision in order to avoid repetition of the biological activities.

Lines 78-79: This sentence must be improved. What are the differences between the studies already performed and reported on E. urophylla leaf oils and the study/results described in the present paper? This should be clearly indicated.

Results and discussion

Lines 119-120please correct the sentence

Lines 177-251: all the sections describing the antibacterial activity of eucalypt oils must be re-written and improved. Several times the authors described the oils as sensitive or extremely sensitive. This is not correct because an oil can’t be sensible to a microorganism. The oil can be active (or effective or have an effect) against a microorganism, and a microorganism can be sensible to the oil. Therefore all this section must be revised and corrected.

Lines 224-226: The different susceptibilities of Gram-positive and Gram-negative microorganisms to the oils are due, not only to a different composition but also can be related to cell wall structural differences between gram-negative and gram-positive bacteria Please discuss accordingly.

Lines 227-250: this part of the text is very confusing and must be summarized.

Lines 273-276: a moderate activity or weak activity is described, comparing with what?

Lines 293-300: this paragraph is not comprehensible. Please revise.

Materials and Methods

What was the positive control used in the FRAP assay?

Conclusions

The last sentence (lines 431-434) must be improved to avoid repetition.

Round 2

Reviewer 1 Report

Dear Editor,

I appreciate an effort of the authors to improve my comments and advise. However, there are some things that still need to be improved. This article is broadly considered for publication in Molecules based on the variation of data in the Antibacterial and Antioxidant Activities of Essential Oils of Five New Eucalyptus urophylla S.T. Blake 3 Clones in Thailand. The data provides new information, especially regarding the Comparison of essential oil yields, main terpenoids, and antibacterial and antioxidant activities of E. urophylla leaf oils with some of those of previous studies. This can also be a consideration for researchers in accelerating the discovery and development of Antibacterial and Antioxidant Activities. Overall, this paper is very interesting. However, there are some things that need to be emphasized and improved, such as the following suggestions:

Major:

  1. Introduction (Page 1, Line 42): said “Eucalyptus is one of Thailand's most important economic woods for pulp and paper production. Two of the most promising Eucalyptus 43 species widely planted in Thailand for commercial forest plantations are E. urophylla S.T. Blake (Timor White Gum, Timor Mountain Gum). It will be better if it is explained that the economic growth data from eucalyptus utilization and data on the percentage of Eucalyptus planting in Thailand.
  2. Introduction (Page 2, Line 61): Hydrodistillation and steam distillation techniques are commonly used to extract essential oils from Eucalyptus leaves. It will be better if it is explained why this technique is widely used now.
  3. Introduction (Page 2, Line 80) Interestingly, E. urophylla and E. camaldulensis oils have been shown to have significant antibacterial effects on a variety of pathogenic Gram-positive and Gram-negative bacteria, whereas the antioxidant properties of E. urophylla oil have lower activity than E. camaldulensis oil. It will be better if it is explained how big is the significance of the data antibacterial effects and the data of the antioxidant properties of E. urophylla oil have lower activity than E. camaldulensis oil

Minor:

  1. Result and discussion (page 3, line 111) the leaf essential oil yields of the E. urophylla and the E. urophylla x E. camaldulensis hybrid are higher than those reported by Li et al. It will be better to add the value urophylla and the E. urophylla x E. camaldulensis hybrid are higher than those reported by Li et al.
  2. Result and discussion (page 3, line 112) The genetic variations in E. camaldulensis leaf essential oil yields have also been reported by our group [20], and the oil yields were higher than E. urophylla oils. It will be better to add the value of the oil yields that were higher than urophylla oils.
  3. Result and discussion (page 7, line 204) Nonetheless, all eucalypt oils were less effective against S. aureus, B. cereus, L. monocytogenes, E. coli, S. typhi, and E. aerogenes than tetracycline, the positive control. It will be better to explain why choose tetracycline as positive control.
  4. Result and discussion (page 10, line 280) The percent inhibition of these oils ranged from 22.56 ±16 to 59.14 ± 1.66 based on the dose of 5.0 mg/mL. It will be better the sentences edited to 22.56 ± 1.16 to 59.14 ± 1.66 %
  5. Materials and Methods (page 11, Line 330) After a week of air drying, the dry leaves of the Eucalyptus samples were chopped into small pieces and ground with blenders. It will be better the sentences edited to the dry leaves of the Eucalyptus samples were chopped into small pieces and ground with blenders after a week of air drying.
  6. Results: there are still things that need to be emphasized and clarified again, such as there is data on n-alkanes in table 1, but there is no explanation regarding the reasons for using n-alkanes and their effect on the chromatogram obtained

Reviewer 2 Report

The authors have revised and improved the manuscript taking into account all the suggestions. However,  English language and style requires a final check.

Author Response

Dear Reviewer,

         Thank you very much for your kind consideration. Anyway, I have already rechecked and edited again after my manuscript has been edited for the English language and style by MDPI English editing services.

Best regards,

Dr. Suwaporn Luangkamin